# GENERALIZATION BOUNDS AND ALGORITHMS FOR ESTIMATING THE EFFECT OF MULTIPLE TREATMENTS AND DOSAGE

## ABSTRACT

Estimating conditional treatment effects has been a longstanding challenge for fields of study such as epidemiology or economics that require a treatment-dosage pair to make decisions, but may not be able to run randomized trials to precisely quantify their effect. This may be due to financial restrictions or ethical considerations. In the context of representation learning, there is an extensive literature relating model architectures with regularization techniques to solve this problem using observational data. However, theoretically motivated loss functions and bounds on generalization errors only exist in selected circumstances, such as in the presence of binary treatments. In this paper, we introduce new bounds on the counterfactual generalization error in the context of multiple treatments and continuous dosage parameters, which subsume existing results. This result, in a principled manner, guides the definition of new learning objectives that can be used to train representation learning algorithms. We show empirically new state-of-the-art performance results across several benchmark datasets for this problem, including in comparison to doubly-robust estimation methods.

## 1 INTRODUCTION

Treatment effect estimation is the problem of predicting the effect of an intervention (e.g. a treatment-dosage pair) on an outcome of interest to guide decision-making. The challenge for prediction models is to learn this map from observational data, which is formally generated from a different structural causal model in which treatment assignment varies according to an individual's covariates, instead of being fixed by the decision-maker. Counterfactuals define the outcome that would have been observed had the assigned treatment been different. For concreteness, consider designing a policy for the administration of chemotherapy regiments; not all cancer patients in the available data are equally likely to be offered the same type and dosage, with varied factors, e.g. age, wealth, etc., involved in the decision-making process. Evaluating a new treatment combination for a given patient is a data point that is invariably under-represented in the empirical distribution of the data.

Treatment effect estimation is studied under a wide range of assumptions, including experimental designs that feature ignorability (Imbens, 2000; Imai & Van Dyk, 2004), multiple treatments, sequential decision-making problems, and different generative models encoded in general causal graphs (Pearl, 2009). There is a growing literature on several parts of this problem in the field of machine learning that attempts to define loss functions that are conducive to learning representations of covariates predictive of both observed and counterfactual outcomes. Existing methods could be generally categorized by the theoretical guarantees that inspire training objectives, driven either by bounds for the generalization error or by doubly-robustness guarantees. In the first line of research, Shalit et al. (2017); Johansson et al. (2020) showed in the binary treatment setting that the counterfactual error, that is not computable from data by design, could be instead bounded by the in-sample error plus a term that quantifies the difference in distributions between treated and untreated populations, leading to a differentiable loss function that can be used to train expressive neural networks. Several papers used this insight to investigate different neural network architectures for this problem. For example, Johansson et al. (2016) proposed to use separate feed-forward prediction heads on top of a common representation, Zhang et al. (2022) use transformers, De Brouwer et al. (2022); Seedat et al. (2022) use neural differential equations. In turn, doubly-robust estimators combine expressive function

approximators and inverse probability weighting leveraging statistical non-parametric asymptotic guarantees of both estimators (Funk et al., 2011; Kennedy, 2016; 2020). In particular, when the direct estimate of the outcome is biased, such as when using nonparametric or high-dimensional regression, the doubly robust estimator weights the model residuals by inverse propensity weights in order to remove the bias. Its convergence and consistency for treatment effect estimation requires only that one of the estimators is consistent. In principle, any consistent function approximator could be used, which in the context of neural networks has led to several adaptations of loss functions and architectures. For example, Shi et al. (2019) adapted the architecture of Johansson et al. (2016) for this purpose introducing targeted regularization, and Nie et al. (2020) proposed varying coefficient networks in the context of continuously-valued dosage parameters. In both cases, however, the authors provide guarantees for population average treatment effect estimation, in contrast with conditional average treatment effect estimation.

Despite the generality of these results, no guarantees and no theoretically motivated loss functions exist for learning representations for counterfactual estimation in the general setting of multiple treatment types and/or continuous treatment values or dosages. The challenge in the context of representation learning is that there is no notion of treatment group as each individual gets assigned a potentially different and unique treatment value. Lack of overlap in finite samples and subsequently large estimation variance for counterfactual predictions are exacerbated in this setting to the extreme that adjustments for distributional differences are, in principle, not applicable. In particular, the intuition for reducing variance by regularization deviates from previous proposals (that regularize representations of covariates to match distributions among groups with different treatment types (Shalit et al., 2017)) as a potentially infinite set of counterfactuals for each individual must be considered. Even the analysis of multiple categorical treatments is currently an open question as, while pairwise comparisons between treatment specific distributions could be implemented in principle, it is not computationally tractable to do so in practice. At this moment, only heuristic neural network architectures for this problem have been proposed, including Dose Response networks that consist of multi-task layers for dosage sub-intervals defined on top of a common representation (Schwab et al., 2020), variants of generative adversarial networks (Bica et al., 2020), and varying coefficient networks (Nie et al., 2020).

In this paper, we investigate the design of representation learning-based algorithms for predicting (conditional average) treatment effects in the context of multiple treatments and continuous dosage parameters. Our analysis starts by extending definitions of loss and generalization error to this broader setting, over all possible treatment-dosage pairs. We then show by using the definition of integral probability metrics that the generalization error can be bounded by a term that is computable from data and that involves the factual error and a term that quantifies the statistical dependence between the pair of treatment-dosage random variables and observed confounders. In principle, any treatment space on which we can define a probability measure is consistently accounted for, which gives well-defined bounds on the generalization error for treatments with multiple types and continuous values, and in particular, our bound includes as a special case existing guarantees in the binary treatment case (Shalit et al., 2017). This bound suggests new training objectives for learning representations conducive to counterfactual estimation. Moreover, such objectives are tractable: both avoiding combinatorial numbers of pairwise comparisons and avoiding binning dosage values into different sub-intervals. A further contribution we make is to design extensive numerical comparisons that compare both methods driven by bounds on the generalization error (that typically target conditional average treatment effects) and methods driven by doubly-robust guarantees (that typically target average treatment effects). Moreover, we do so independently of the adopted neural architecture which provides the first analysis of different objectives for the problem of treatment effect estimation with multiple, continuously-valued treatments. We hope these results can give some insight into the trade-offs of different approaches to this problem and demonstrate the ability of representation learning techniques to tackle wider ranging scenarios within treatment effect estimation.

## 2 BACKGROUND

We start by introducing the notation and definitions used throughout the paper. In particular, we use capital letters for random variables ($X$), small letters for their values ($x$), bold letters for sets of variables ($\mathbf{X}$) and their values ($\mathbf{x}$), and $\Omega$ for the spaces where they are defined ($\Omega_{\mathbf{X}}$) if not explicitly stated. To simplify notation, we consistently use the shorthand $P(\mathbf{x})$ to represent probabilities or

densities $P(\mathbf{X} = \mathbf{x})$ and similarly $P(y \mid \mathbf{x})$ to represent $P(Y = y \mid \mathbf{X} = \mathbf{x})$. For three sets of variables $\mathbf{X}, \mathbf{Y}, \mathbf{Z}$ the conditional independence statement "$\mathbf{X}$ is conditionally independent of $\mathbf{Y}$ given $\mathbf{Z} = \mathbf{z}$" is written as $\mathbf{X} \perp\!\!\!\perp \mathbf{Y} | \mathbf{Z}$.

We use the semantics of the Rubin-Neyman potential outcomes framework, see e.g. Section 2 in Rubin (2005). We assume that for an individual with observed covariates $\mathbf{x} \in \Omega_{\mathbf{X}}$, and tuple $T = (W, S)$ defining the treatment type out of $k$ distinct treatments $W \in \Omega_W = \{w_1, \dots, w_k\}$ and dosage parameter $S \in \Omega_S = \mathbb{R}$, there is a corresponding potential outcome $Y_t$ that would have been observed had the assigned treatment been $T = t$. With observational data only one of these potential outcomes is observed for each unit depending on the treatment assignment. We will refer to the unobserved potential outcomes as counterfactuals. Let $Y$ denote the observed outcome and $\Omega_T = \{(w, s) : w \in \Omega_W, s \in \mathbb{R}\}$ denote the set of all treatment options. The goal is to derive estimates of the expected potential outcomes for a given set of input covariates: $\mathbb{E}[Y_t \mid \mathbf{x}]$, for any value of $t$ and $\mathbf{x}$. Under the following standard assumptions (Rubin, 2005), it is well understood that the treatment effect between two selected treatment options $t_1$ and $t_2$ reduces to a contrast of conditional distributions, presented in Prop. 1 below.

**Assumption 1** (Unconfoundedness). *The treatment assignment, $t = (w, s) \in \Omega_T$, and potential outcomes, $Y_t$, are conditionally independent given the covariates $\mathbf{x}$, i.e. $Y_T \perp\!\!\!\perp T \mid \mathbf{X}$.*

**Assumption 2** (Overlap). *For any $\mathbf{x} \in \Omega_X$ such that $P(\mathbf{x}) > 0$, we have $1 > P(t \mid \mathbf{x}) > 0$ for each $t \in \Omega_T$.*

**Assumption 3** (Consistency). *The observed outcome is the potential outcome, as a function of treatment, when the treatment is set to the observed exposure, i.e. $Y = Y_t$ if $T = t$ for any $t \in \Omega_T$.*

**Proposition 1** (Identifiability). *Under assumptions 1 and 3, and any $t_1, t_2 \in \Omega_T$,*

$$\mathbb{E}[Y_{t_1} - Y_{t_2} \mid \mathbf{x}] = \mathbb{E}[Y \mid \mathbf{x}, t_1] - \mathbb{E}[Y \mid \mathbf{x}, t_2], \tag{1}$$

*which is composed entirely of observable quantities and can be estimated from data given Assump. 2.*

We refer to the quantity $\mathbb{E}[Y_{t_1} - Y_{t_2} \mid \mathbf{x}]$ as the conditional (or individual if the conditioning set identifies a unit) treatment effect (CATE), and the $\mathbb{E}[Y_{t_1} - Y_{t_2}]$ as the average, or population, treatment effect (ATE). Our results rely on defining representation functions $\phi : \Omega_{\mathbf{X}} \to \Omega_{\mathbf{R}}$, where $\Omega_{\mathbf{R}}$ is the representation space, that preserve unconfoundedness and overlap, and the identifiability of the treatment effect. For this purpose, it is sufficient to assume $\phi$ to be injective[1].

**Corollary 1** (Identifiability given representation). *Under the assumption that the representation $\phi$ is injective, $1 > P(t \mid \phi(\mathbf{x})) > 0$ and $Y_T \perp\!\!\!\perp T \mid \phi(\mathbf{X})$, that is unconfoundedness and overlap hold conditional on features $\phi(\mathbf{x})$.*

Without loss of generality we will assume that $\Omega_{\mathbf{R}}$ is the image of $\Omega_{\mathbf{X}}$ under $\phi$. We will write $P$ also to denote the distribution induced by $\phi$ over $\Omega_{\mathbf{R}}$ and let $h : \Omega_{\mathbf{R}} \times \Omega_T \to \Omega_Y$ be a prediction function defined over $\Omega_{\mathbf{R}}$. Next, we define two complimentary loss functions: one is the standard machine learning loss, which we will call the factual error on the estimation at the observed treatment type and dosage tuple, and the other is the counterfactual error, as an average error over all other treatment assignment options, made for an individual with a particular treatment type and dosage tuple.

**Definition 1.** For a given loss function $\mathcal{L} : \Omega_Y \times \Omega_Y \to \mathbb{R}^+$, the expected factual and counterfactual losses of $h$ and $\phi$ at treatment $t \in \Omega_T$ are defined as,

$$\mathcal{L}_F(t) = \int_{\Omega_X} \int_{\Omega_Y} \mathcal{L}(y_t, h(\phi(\mathbf{x}), t)) P(y_t|\mathbf{x}) P(\mathbf{x}|t) d\mathbf{x} dy_t, \tag{2}$$

$$\mathcal{L}_{CF}(t) = \mathbb{E}_{t' \sim P} \int_{\Omega_X} \int_{\Omega_Y} \mathcal{L}(y_t, h(\phi(\mathbf{x}), t)) P(y_t|\mathbf{x}) P(\mathbf{x}|t') d\mathbf{x} dy_t. \tag{3}$$

The counterfactual error defines the average error made for the counterfactual prediction at treatment tuple $t = (w, s)$ on all individuals that are observed to be assigned a different treatment $t' \neq t$. This

---

[1]The remark has been made that injectivity of representation is difficult to enforce (Zhang et al., 2020; Johansson et al., 2019). An algorithmic solution, discussed by Zhang et al. (2020), is to include a decoder from the representation to the input domain and reconstruction loss in the training objective to encourage solutions with invertible latent representations. A reconstruction loss and encoder-decoder architecture can be included on top of the regularization terms proposed in this paper.

definition extends the binary treatment case to assess the quality of counterfactual predictions at $t \in \Omega_T$. Similarly, we define an average measure of factual and counterfactual performance over all possible treatment options $t \in \Omega_T$.

**Definition 2.** The average factual and counterfactual error over all treatment options are defined.

$$\mathcal{L}_F = \int_{\Omega_T} \mathcal{L}_F(t)P(t)dt, \quad \mathcal{L}_{CF} = \int_{\Omega_T} \mathcal{L}_{CF}(t)P(t)dt. \tag{4}$$

Next we define the error made on the estimation of a counterfactual contrast for a given pair of treatments, instead of an average over all counterfactual treatment options.

**Definition 3.** Let the treatment effect between two different treatments tuples $t_1, t_2 \in \Omega_T$ be given by $\tau_{(t_1,t_2)}(\mathbf{x}) = \mathbb{E}[Y_{t_1} \mid \mathbf{x}] - \mathbb{E}[Y_{t_2} \mid \mathbf{x}]$. The error in treatment effect estimation is then defined as,

$$\mathcal{L}_{(t_1,t_2)} := \int_{\Omega_{\mathbf{X}}} \mathcal{L}(\tau_{(t_1,t_2)}(\mathbf{x}), \hat{\tau}_{(t_1,t_2)}(\mathbf{x}))P(\mathbf{x})d\mathbf{x}, \tag{5}$$

where $\hat{\tau}_{(t_1,t_2)} : \Omega_{\mathbf{X}} \to \Omega_Y$ denotes its estimate.

## 3 REPRESENTATION LEARNING FOR COUNTERFACTUAL ESTIMATION

As is apparent in the presence of multiple treatments and continuously-valued dosages, there is no notion of treatment group as each individual gets assigned a potentially different and unique treatment value. The intuition for reducing variance by regularization deviates from previous proposals as a potentially infinite set of counterfactuals for each individual must be considered Shalit et al. (2017). The following theorem shows that the average counterfactual error defined in Def. 2 can be bounded by terms that are explicitly computable from observational data.

**Theorem 1** (Bound on average counterfactual generalization error). *Under the assumption that $\phi$ is injective, it holds that,*

$$\mathcal{L}_{CF} \leqslant \mathcal{L}_F + \lambda \cdot \sup_{g \in \Omega_g} \Big| \int_{\Omega_T} \int_{\Omega_{\mathbf{R}}} g(\mathbf{r}, t) \cdot (P(\mathbf{r})P(t) - P(\mathbf{r}, t))d\mathbf{r}dt \Big|. \tag{6}$$

$\Omega_g$ *defines a space of functions* $g : \Omega_{\mathbf{R}} \times \Omega_T \to \mathbb{R}$ *expressive enough to include* $\int_{\Omega_Y} \mathcal{L}(y_t, h(\phi(\mathbf{x}), t))P(y_t|\mathbf{x})dy_t/\lambda$ *as a function of $\phi(\mathbf{x})$ and $t$, where $\lambda > 0$ depends on the choice of representation function $\phi$.*

This theorem states that the average counterfactual error is upper-bounded by the factual error plus a term that quantifies the dependence between treatment tuple $T$ and covariates $\mathbf{X}$. As the treatment tuple contains multiple treatment types $w$, as well as continuous dosages $s$, this single bound is valid for multiple treatment values as well as continuous dosages.

**Bias Variance tradeoff** Counterfactual estimation would be unbiased by minimizing factual losses $\mathcal{L}_F$ by Prop. 1, but the variance in the estimation of counterfactuals for treatment-dosage pairs that are not heavily represented in observational data will be high. This will contribute to larger generalization error and is captured in the supremum in the second term of Eq. (6). In particular, the supremum quantifies an imbalance in the association of $T$ and $\mathbf{R}$ by using distributional distances between joint distributions and the product of marginals. $|P(\mathbf{r})P(t) - P(\mathbf{r}, t)|$ is large if not all treatment and feature combinations are evenly represented in the data. This observation recovers an interesting intuition if $\Omega_g$ is chosen to be expressive enough. The observation being that $\sup_{g \in \Omega_g} \Big| \int_{\Omega_T} \int_{\Omega_{\mathbf{R}}} g(\mathbf{r}, t) \cdot (P(\mathbf{r})P(t) - P(\mathbf{r}, t))d\mathbf{r}dt \Big| = 0$ if and only if the representation is independent of treatment assignment, i.e. $\phi(\mathbf{X}) \perp\!\!\!\perp T$. This extreme case leads to lower variance as counterfactuals for a treatment-dosage tuple have the same effective sample size as that of the observational data. The hyperparamter $\lambda$ controls the tradeoff between the bias and the variance of the counterfactuals. Two choices for $\Omega_g$ we consider are the space of functions in a universal Reproducing Kernel Hilbert Space (RKHS) with characteristic kernels (Sriperumbudur et al., 2011), which recovers the well-known Hilbert Schmidt Independence Criterion (Gretton et al., 2007), and the space of Lipschitz functions with Lipschitz constant bounded by 1 which recovers the Wasserstein distance (Villani, 2009).

**Binary treatment case** One insight from Thm. 1 is that bias in the treatment assignment in the context of a general treatment choices, such as multiple treatment types or continuously-valued treatments, takes the form of high statistical dependence between random variables, that is more general than differences between distributions. In particular, differences in distributions between treatment groups as defined by Shalit et al. (2017) in the binary treatment case can be formulated as statistical dependence between random variables. The following corollary recovers the generalization bound of Shalit et al. (2017) as a special case.

**Corollary 2.** *Let* $\Omega_T = \{0, 1\}$. *Then, by Thm. 1,*

$$\mathcal{L}_{CF} \leqslant \mathcal{L}_F + \lambda \cdot \sup_{g \in \Omega_g} \Big| \int_{\Omega_{\mathbf{R}}} g(\mathbf{r}) \cdot (P(\mathbf{r} \mid T = 1) - P(\mathbf{r} \mid T = 0)) d\mathbf{r} \Big|, \tag{7}$$

*and is equivalent to (Shalit et al., 2017, Lemma 1).*

We show next a similar result that gives generalization bounds for the treatment effect comparing two specific treatment options, instead of an average over all possible counterfactual options, that may be of interest in applications specifically comparing two treatment options.

**Theorem 2.** *Let* $t_1, t_2 \in \Omega_T$ *be two treatment tuples to be compared. Then,*

$$\mathcal{L}_{(t_1, t_2)}/2 \leqslant \mathcal{L}_F(t_1) + \sup_{g \in \Omega_g} \Big| \int_{\Omega_{\mathbf{R}}} g(\mathbf{r}) \cdot (P(\mathbf{r}) - P(\mathbf{r} \mid T = t_1)) d\mathbf{r} \Big| + \mathcal{L}_F(t_2)$$

$$+ \sup_{g \in \Omega_g} \Big| \int_{\Omega_{\mathbf{R}}} g(\mathbf{r}) \cdot (P(\mathbf{r}) - P(\mathbf{r} \mid T = t_2)) d\mathbf{r} \Big| - \sigma_{Y_{t_1}} - \sigma_{Y_{t_2}}, \tag{8}$$

*where* $\sigma_{Y_{t_1}}$ *and* $\sigma_{Y_{t_2}}$ *stand for the variance of the random variables* $Y_{t_1}$ *and* $Y_{t_2}$, *respectively, under the distribution* $P(\mathbf{x})$.

### 3.1 ARCHITECTURES AND ALGORITHMS FOR COUNTERFACTUAL ESTIMATION

This section discusses the architectures of the representation and prediction functions used, as well as training objectives to leverage the generalization bound in Thm. 1. The training objective that we define can be used with any neural network architecture that parameterizes a representation function $\phi_\eta : \Omega_{\mathbf{X}} \to \Omega_{\mathbf{R}}$ and a separate prediction function $h_\theta : \Omega_{\mathbf{R}} \times \Omega_T \to \Omega_Y$ with sets of parameters $\eta$ and $\theta$ respectively.

Following the discussion in Sec. 3, we learn a representation $\phi$ and prediction function $h$ to minimize a trade-off between predictive accuracy and imbalance in the representation space using the following objective:

$$\min_{\theta, \eta} \sum_{n=1}^{N} \left( y^{(n)} - h_\theta(\phi_\eta(\mathbf{x}^{(n)}), t^{(n)}) \right)^2 + \gamma \cdot \text{IPM}_{\Omega_g}(\phi(\mathbf{X}), T), \tag{9}$$

where $\gamma \geqslant 0$ is a hyperparameter, $n$ is the number of samples, and $\text{IPM}_{\Omega_g}(\phi(\mathbf{X}), T) := \sup_{g \in \Omega_g} \left| \int_{\Omega_T} \int_{\Omega_{\mathbf{R}}} g(\mathbf{r}, t) \cdot (P(\mathbf{r})P(t) - P(\mathbf{r}, t)) d\mathbf{r} dt \right|$ is the integral probability metric for a chosen space of functions $\Omega_g$.

Concretely, we wish to increase the predictive accuracy while making the representation as independent of the treatment as possible. We consider the Hilbert Schmidt Independence Criterion (HSIC) and the Wasserstein distance as choices for the integral probability metric. In practice, the HSIC can be approximated with a finite data sample using (Gretton et al., 2007, Eq. (3)). For the Wasserstein distance, we simulate a sample with joint distribution $P(\mathbf{r})P(t)$ by randomly permuting the observed treatment-dosage pair across individuals to generate a sample $\{(\mathbf{r}^{(n)}, t^{(\sigma(n))}) : n = 1, \dots, N\}$, where $\sigma : \{1, \dots, N\} \to \{1, \dots, N\}$ is a bijective function. The original data $\{(\mathbf{r}^{(n)}, t^{(n)}) : n = 1, \dots, N\}$ is drawn from the distribution $P(\mathbf{r}, t)$. The two empirical distributions are compared using the arguments in (Cuturi & Doucet, 2014). Both these regularization terms are differentiable and all parameters can be updated using stochastic gradient descent.

Each treatment type $w$ corresponds to a separate prediction network head, i.e. $h_\theta := \{h_\theta^{(w)}\}_{w \in \Omega_W}$, while the representation layer is common across all treatment types. In particular, this implies that each

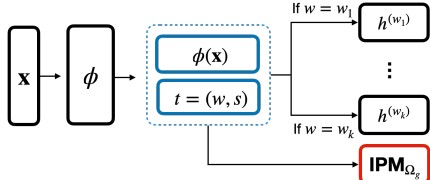

Figure 1: Sketch of the architecture.

sample $(\mathbf{x}^{(n)}, w^{(n)}, s^{(n)}, y^{(n)})$ is used to update only the prediction network $h_\theta^{(w^{(n)})}$ corresponding to the observed treatment $w^{(n)}$, while all samples are used to update the representation layer $\phi_\eta$. A sketch of this training routine is given in Fig. 1.

The following network architectures for the prediction functions $\{h^{(w)}\}_{w \in \Omega_W}$ have been proposed in the literature.

**Dose Response Networks (DRNet)**   Schwab et al. (2020) propose Dose Response Networks for predicting the effect of dosage on an outcome of interest. The architecture takes the form of a multi-task network with a shared set of layers and multiple task-specific heads. In this context, the range of dosage values is split into separate bins and each of them is associated with a separate head. Each task-specific network in addition takes the dosage value as input, but crucially the parameterization of the prediction function is common to all dosages belonging to the same sub-interval. For example, the range of dosage values for treatment type $w$ could be divided into 5 sub-intervals, thus using 5 task-specific heads $h_\theta^{(w)} = (h_\theta^{(w,1)}, \ldots, h_\theta^{(w,5)})$, $h_\theta^{(w,i)} : \Omega_{\mathbf{R}} \times \Omega_T \to \Omega_Y, i = 1, \ldots, 5$. To some extent this approach accounts for the heterogeneity in the dose-response function but remains limited by the binning choice and may be vulnerable to abrupt changes in the prediction on the dosage values that separate two bins, as demonstrated by Nie et al. (2020).

**Varying Coefficient networks (VCNet)**   Varying Coefficient networks (Nie et al., 2020) are proposed for dose-response estimation, but a multi-task architecture can be designed as a special case. In particular, the authors define the parameters $\theta$ for each prediction network $h_\theta^{(w)} := h_{\theta(s)}^{(w)} : \Omega_{\mathbf{R}} \to \Omega_Y$ to be functions $\theta(s) = (\theta_1(s), \ldots, \theta_{d_\theta}(s))$ of dosage themselves, where $d_\theta$ is the total number of parameters. Each scalar parameter $\theta_i : \Omega_S \to \mathbb{R}$ is given by a linear combination $\theta_i(s) = \sum_{l=1}^L \alpha_{i,l} \psi_l(s)$ of polynomial basis functions $\{\psi_l\}_{l=1}^L$ defined on the space of dosage values $\Omega_S$. The coefficients $\{\alpha_{i,l} : i = 1, \ldots, d_\theta, l = 1, \ldots, L\}$ define the trainable parameters and the map $h_\theta^{(w)}(\mathbf{r}, s) := h_{\theta(s)}^{(w)}(\mathbf{r}, s)$ is differentiable with respect to $\{\alpha_{i,l} : i = 1, \ldots, d_\theta, l = 1, \ldots, L\}$. For example, DRNets are recovered by choosing $\{\psi_l\}_{l=1}^L$ to be a piece-wise constant functions spline basis of the form $\mathbb{1}(s_i \leqslant t < s_j)$ with different $s_i, s_j$. More general choices can be made, such as B-splines, that lead to continuous dose response curves. The influence of the dosage parameter is different to that of a covariate and thus ensures dosage information is not lost in high-dimensional representations, which in practice has been shown to lead to better counterfactual prediction performance.

## 4   EXPERIMENTS

This section conducts controlled experiments on synthetic and semi-synthetic datasets previously used in the literature. Overall, we found that simulation results support our generalization guarantees with different architectures benefiting from the proposed regularization strategy using both the HSIC and Wasserstein distances.

### 4.1   BASELINES AND METRICS

We consider several baselines for comparison, including different neural network architectures without regularization and with doubly-robust regularization techniques. In particular, we consider a standard multilayer perceptron (MLP) that optimises the (factual) squared error loss objective to learn the weights of the network, a standard VCNet (Nie et al., 2020), and DRNet (Schwab et al., 2020). In

the context of doubly-robust optimization, Shi et al. (2019); Nie et al. (2020) propose to learn a joint representation $\phi(\mathbf{x})$ that is conducive to both counterfactual $h_1 : \Omega_\mathbf{R} \times \Omega_T \to \Omega_Y$ and propensity score estimation $h_2 : \Omega_\mathbf{R} \to \Omega_T$ by a using a loss function that trades-off the two objectives, e.g.,

$$\frac{1}{N} \sum_{n=1}^{N} \left( y^{(n)} - h_1(\phi(\mathbf{x}^{(n)}), t^{(n)}) \right)^2 + \alpha \cdot \text{CrossEntropy}\left( h_2(\phi(\mathbf{x}^{(n)})), t^{(n)} \right), \qquad (10)$$

If $h_1$ and $h_2$ are consistent estimators of the outcome and propensity scores respectively, as well as satisfy the non-parametric estimating equation,

$$\frac{1}{N} \sum_{n=1}^{N} \mu(y^{(n)}, t^{(n)}, \mathbf{x}^{(n)}; \hat{h}_1, \hat{h}_2, \hat{\epsilon}) = 0, \qquad (11)$$

where $\epsilon$ denotes a perturbation term that is optimized and where (in the binary treatment case for simplicity),

$$\mu(y, t, \mathbf{x}; h_1, h_2, \epsilon) = h_1(\mathbf{x}, 1) - h_1(\mathbf{x}, 0) + \left( \frac{t}{h_2(\mathbf{x})} - \frac{1-t}{1-h_2(\mathbf{x})} \right) \cdot (y - h_1(\mathbf{x}, t)) - \epsilon, \quad (12)$$

then the resulting estimator will have desirable asymptotic properties for average treatment effect (Shi et al., 2019; Kennedy, 2016). We consider $h_1$ parameterized by both VCNets and DRNets. Algorithms trained to minimize Eq. (10) are denoted VCNet-PS, DRNet-PS, and algorithms trained to minimize both Eqs. (10) and (11) (also known as Targeted Regularization), are referred to as VCNet-TR, DRNet-TR. Finally, we consider Generalized Propensity Scores (GPS) (Imbens, 2000; Imai & Van Dyk, 2004) that fit a linear model using inverse propensity scores. Our proposed methods are labeled DRNet-HSIC, DRNet-Wass, VCNet-HSIC, and VCNet-Wass, which combine existing architectures with the proposed regularization methods. We include details on network architectures, hyperparameters optimisation and computational time in Appendix C.

For performance comparisons, we consider the Mean Integrated Squared Error (MISE),

$$\text{MISE} = \frac{1}{N} \frac{1}{k} \sum_{n=1}^{N} \sum_{w \in \Omega_W} \mathbb{E}\left[ \left( y_{(w,s)}^{(n)} - \hat{y}_{(w,s)}^{(n)} \right)^2 \right], \qquad (13)$$

where we use the notation $y_{(w,s)}^{(n)}$ and $\hat{y}_{(w,s)}^{(n)}$ for the true and predicted outcome for individual $n$ given treatment-dosage pairs $(w, s) \in \Omega_T$, and the expectation is taken with respect to the dosage parameter, i.e. $\mathbb{E}\left[ y_{(w,s)}^{(n)} \right] = \int_{\Omega_S} y_{(w,s)}^{(n)} P(s) ds$. Intuitively, MISE calculates how well an algorithm is at estimating individual level dose response and thus accounts for the heterogeneity in treatment response. In contrast, the Average Mean Squared Error (AMSE) evaluates population average counterfactual prediction by taking sums and integrals before comparisons between predicted and true outcomes. We define and evaluate AMSE in Appendix D.

## 4.2 DATASETS

The nature of the treatment-effects estimation problem does not allow for meaningful evaluation on real-world datasets. This is simply because we never observe a counterfactual for a given unit. There are, however, established synthetic and semi-synthetic datasets that have been used by Schwab et al. (2020); Bica et al. (2020); Nie et al. (2020). Following these proposals we use,

- *Fully synthetic.* A data generating mechanism with a total of 6 randomly generated covariates and a single treatment with dosage ranging from 0 to 1 that involve complex functions for both treatment assignment and outcome function, as defined by Nie et al. (2020).
- *IHDP-continuous.* The original semi-synthetic IHDP dataset from Hill (2011) contains binary treatments with 747 observations on 25 covariates. We adapt this dataset to the continuous dosage context by changing the treatment assignment and outcome function. We generate these in a similar way to Nie et al. (2020).
- *News.* The News dataset consists of 3000 randomly sampled news items from the NY Times corpus (Newman, 2008), which was originally introduced as a benchmark in the binary treatment setting. We generate a continuously-valued treatment and corresponding outcome in a similar way as Bica et al. (2020).

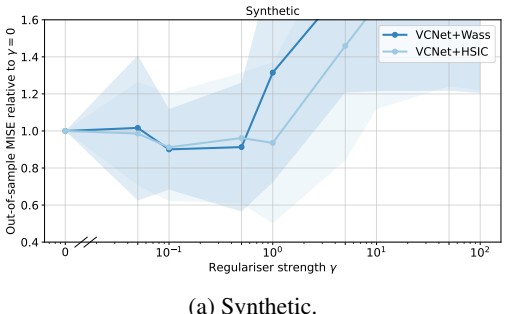 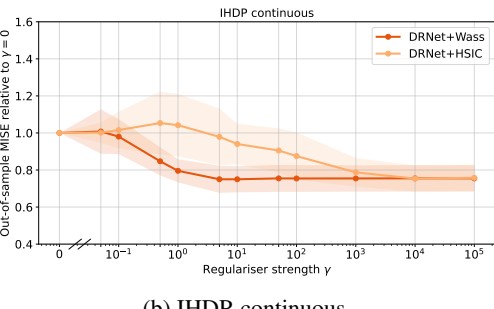

(a) Synthetic.                    (b) IHDP-continuous.

Figure 2: Out-of-sample MISE error versus IPM regularization, relative to the error at $\gamma = 0$ (no regularisation), on 50 realizations of Synthetic (a) and IHDP-continuous (b) datasets. Average values (dot markers) and one standard deviation (shaded areas) are shown.

In each of our experiments we generate 50 independent realizations from each of the above datasets (20 for News), with samples split into a train/validation/test set with ratios 0.6/0.2/0.2. Further details on the data generating mechanisms, as well as about networks architecture, hyper-parameters tuning and training times are provided in Appendix C.

### 4.3 EFFECTIVENESS OF REGULARISATION

Our first experiment tests the effectiveness of the proposed regulariser by evaluating counterfactual prediction performance as a function of $\gamma$ that determines the influence of the independence constraint in feature space in Eq. (9).

We consider both DRNets and VCNets architectures, with both HSIC and Wasserstein regularizers on the Synthetic and IHDP-continuous datasets. Fig. 2 compares MISE performance results for these models with varying values of $\gamma$ relative to $\gamma = 0$ (without regularisation). Both datasets include confounding factors which induce bias or imbalance in the treatment assignment $T$ for different covariate subgroups $\mathbf{X}$. On both plots we observe that the proposed regularization term (with increasing $\gamma > 0$ relative to $\gamma = 0$) confers an advantage to training with a regularization term that explicitly corrects for this imbalance for the purpose of predicting counterfactuals. The gain of some $\gamma > 0$ is consistent across different neural network architectures and across different datasets, which illustrates our generalization guarantees but also shows that some form of regularisation may broadly be applicable in practice.

### 4.4 PERFORMANCE COMPARISONS

In this section we conduct a wide-range comparison against the benchmark prediction algorithms using the three data generating mechanisms described in Section 4.2. Table 1 reports average values and standard deviations of $\sqrt{\text{MISE}}$ over 50 (20) realizations of Synthetic and IHDP-continuous (News) datasets. On average, the proposed regularization technique, using either the HSIC or Wasserstein distances between distributions, outperforms all other regularization techniques on both choices of neural network architecture. Several trends are interesting to discuss in more detail.

Existing representation learning algorithms that optimise doubly-robust objectives are not always optimal. The results show that, in terms of the MISE, our regularisation based on counterfactual generalisation outperforms doubly robust methods. This can be explained by the fact that doubly robust methods have guarantees when estimating average treatment effects, and not individual or conditional treatment effects. The proposed regularization techniques, with guarantees for counterfactual generalization error, instead, are designed for good performance in conditional average treatment effect estimation and often substantially outperform in terms of MISE. We believe that this discrepancy is due to the doubly robust methods discarding information that helps predict the individual outcome, resulting in a worse MISE performance. This also emphasizes the fact that estimating average counterfactuals and individual counterfactuals can require different objectives. Indeed, the cross-entropy term in Eq. (10) encourages the representation to retain information that is predictive of the treatment; hence, it encourages the discarding of information that is predictive of

|            | **Synthetic** | **IHDP-continuous** | **News**    |
|------------|---------------|---------------------|-------------|
| GPS        | 2.80 (0.51)   | 4.91 (0.87)         | -           |
| MLP        | 0.72 (0.09)   | 0.74 (0.07)         | 1.05 (0.07) |
| DRNet      | 0.34 (0.06)   | 0.60 (0.06)         | 0.84 (0.04) |
| DRNet-PS   | 0.43 (0.07)   | 0.66 (0.15)         | 0.84 (0.04) |
| DRNet-TR   | 0.41 (0.03)   | 2.07 (3.54)         | 0.82 (0.05) |
| DRNet-HSIC | 0.39 (0.06)   | 0.52 (0.05)         | 0.87 (0.04) |
| **DRNet-Wass** | 0.38 (0.07) | **0.51 (0.08)**   | 0.86 (0.04) |
| VCNet      | 0.33 (0.03)   | 0.56 (0.09)         | 0.85 (0.05) |
| VCNet-PS   | 0.62 (0.42)   | 0.59 (0.11)         | 0.98 (0.09) |
| VCNet-TR   | 0.42 (0.12)   | 0.64 (0.39)         | 0.99 (0.06) |
| **VCNet-HSIC** | **0.28 (0.04)** | 0.56 (0.09)   | 0.87 (0.05) |
| **VCNet-Wass** | 0.38 (0.03) | 0.55 (0.10)       | **0.81 (0.05)** |

Table 1: Average values and standard deviations (within brackets) of $\sqrt{\mathrm{MISE}}$ across 50 (20) realizations of Synthetic, IHDP-continuous (News) datasets. Bold notation highlights the best-performing algorithm on each dataset.

the outcome but not the treatment, which is simply noise when predicting the treatment. On average there is also a significant gain by considering more expressive neural network architectures, for instance DRNet outperforms MLP and VCNet outperforms DRNet on all metrics and data generating mechanisms. Finally, we note that GPS requires matrix inversion which was not feasible to compute on the high-dimensional News dataset.

## 5 CONCLUSION

In this paper, we investigate the task of estimating the conditional average causal effect of dosage from a combination of observational data and assumptions on the causal relationships in the underlying system. When these assumptions hold, we give new bounds on the counterfactual generalization error in the context of multiple treatment types and continuously-valued dosage parameters that subsume generalization guarantees from the binary treatment case. Using this result, we provide new learning objectives that can be used to guide the training of representation learning algorithms. We show empirically new state-of-the-art performance results across several benchmark datasets for this problem. To our knowledge, this is the first paper exploring representation learning and regularization for conditional average counterfactual estimation in the context of multiple, continuous-valued treatments in a principled manner. We hope these results can demonstrate the ability of representation learning techniques to tackle wider ranging scenarios within treatment effect estimation.

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

## A    RELATED WORK ON DOUBLY ROBUST ESTIMATION OF THE AVERAGE TREATMENT EFFECT

Thm. 1 suggests that the imbalance in the distribution of $\mathbf{X}$ across treatment dosage pairs is relevant for the expected generalization error of fitted models. Estimators inspired from the semi-parametric literature, known as doubly robust estimators (Van Der Laan & Rubin, 2006; Chernozhukov et al., 2017), instead try to optimize average treatment effects (ATE), e.g. $\mathbb{E}_{\mathbf{X}}\mathbb{E}[Y_1 \mid \mathbf{x}] - \mathbb{E}_{\mathbf{X}}\mathbb{E}[Y_0 \mid \mathbf{x}]$, by constructing a prediction function $h_1 : \Omega_{\mathbf{X}} \times \Omega_T \to \Omega_Y$, a propensity score function $h_2 : \Omega_{\mathbf{X}} \to \Omega_T$, and perturbation term $\epsilon$, satisfying the non-parametric estimating equation,

$$\frac{1}{N} \sum_{n=1}^{N} \mu(y^{(n)}, t^{(n)}, \mathbf{x}^{(n)}; \hat{h}_1, \hat{h}_2, \hat{\epsilon}) = 0, \tag{14}$$

where (in the binary treatment case for simplicity),

$$\mu(y, t, \mathbf{x}; h_1, h_2, \epsilon) = h_1(\mathbf{x}, 1) - h_1(\mathbf{x}, 0) + \left( \frac{t}{h_2(\mathbf{x})} - \frac{1-t}{1 - h_2(\mathbf{x})} \right) \cdot (y - h_1(\mathbf{x}, t)) - \epsilon. \tag{15}$$

$h_1(\mathbf{x}, t)$ is an estimator of $\mathbb{E}[Y_t \mid \mathbf{x}]$, while $h_2(\mathbf{x})$ is an estimator of the probability of treatment $P(t \mid \mathbf{x})$ and $\epsilon \in \Omega_T$ is a perturbation term that is optimized. In the literature, a common estimation approach is to rely on (task-agnostic) fitted models $\hat{h}_1$ and $\hat{h}_2$, and then choose $\epsilon$ so that this equation is satisfied. If $h_1$ and $h_2$ are consistent estimators of the outcome and propensity scores respectively, as well as satisfy Eq. (14), the resulting estimator of the ATE will have desirable asymptotic properties (Shi et al., 2019; Kennedy, 2016). However, as these guarantees are on the average treatment effects, they do not necessarily guarantee accurate estimates of conditional treatment effects.

In the context of neural networks, Shi et al. (2019); Nie et al. (2020) propose to learn a joint representation $\phi(\mathbf{x})$ that is conducive to both counterfactual $h_1 : \Omega_{\mathbf{R}} \times \Omega_T \to \Omega_Y$ and propensity score estimation $h_2 : \Omega_{\mathbf{R}} \to \Omega_T$ by a using a loss function that trades-off the two objectives, e.g.,

$$\frac{1}{N} \sum_{n=1}^{N} \left( y^{(n)} - h_1(\phi(\mathbf{x}^{(n)}), t^{(n)}) \right)^2 + \alpha \cdot \text{CrossEntropy} \left( h_2(\phi(\mathbf{x}^{(n)})), t^{(n)} \right), \tag{16}$$

as in (Nie et al., 2020, Eq. (1)) or (Shi et al., 2019, Eq. (2.2)). The motivation is that: "If the average treatment effect is identifiable conditioning on the propensity score [. . . ] it suffices to adjust for only the information in $\mathbf{x}$ that is relevant for predicting the treatment", see (Shi et al., 2019, Theorem 2.1). Intuitively, the cross entropy term in Eq. (16) encourages the representation to retain information that is predictive of the treatment. Hence, it encourages the discarding of information that is predictive of the outcome but not the treatment, which is simply noise when predicting the treatment.

Variables that affect the outcome and not treatment are referred to as effect modifiers in the literature, see e.g. (Hernán & Robins, 2010). By definition, the treatment effect varies across different conditioning sets of these effect modifiers. As effect modifiers are responsible for the heterogeneity of treatment effects, it is necessary to condition on them to obtain accurate conditional treatment effects. Thus, to compute conditional average or "individualized" treatment effects such representations may be too restrictive because they tend to ignore effect modifiers.

In contrast, our regularizer penalizes the dependence between the representation and the treatment distributions explicitly. Loosely speaking we discard covariate information predictive of treatment but outcome information is retained. Hence, our regularizer should preserve these effect modifiers leading to more accurate estimates of conditional treatment effects. We conclude that, in general, optimal average treatment effects does not necessarily imply optimal conditional average treatment effects as measured by expected losses in Definitions 1 and 2[2]. We verify this intuition in our experiments.

---

[2]Definitions 1 and 2 also involve averages but makes a head to head comparisons between observed outcomes and predicted outcomes for each individual $(\mathbf{x}, t)$ in the term $\int_{\Omega_Y} \mathcal{L}(y_t, h(\phi(\mathbf{x}), t)) P(y_t | \mathbf{x}) dy_t$ (which are then averaged across individuals) instead of averaging predicted counterfactuals across the whole population before comparison with average true outcomes across different dosage levels.

## B    PROOFS

**Theorem 1** (Generalization bound for the average counterfactual error). *Under the assumption that $\phi$ is one to one, it holds that,*

$$\mathcal{L}_{CF} \leqslant \mathcal{L}_F + \lambda \cdot \sup_{g \in \Omega_g} \Big| \int_{\Omega_T} \int_{\Omega_{\mathbf{R}}} g(\mathbf{r}, t) \cdot (P(\mathbf{r})P(t) - P(\mathbf{r}, t)) d\mathbf{r} dt \Big|. \tag{17}$$

$\Omega_g$ *defines a space of functions $g : \Omega_{\mathbf{R}} \times \Omega_T \to \mathbb{R}$ expressive enough to include $\int_{\Omega_Y} \mathcal{L}(y_t, h(\phi(\mathbf{x}), t))P(y_t|\mathbf{x})dy_t/\lambda$ as a function of $\phi(\mathbf{x})$ and $t$, where $\lambda > 0$ depends on the choice of representation function $\phi$.*

*Proof.* Let $\psi : \Omega_{\mathbf{R}} \to \Omega_{\mathbf{X}}$ be the inverse of $\phi$ and let $l_{h,\phi}(\mathbf{x}, t) := \int_{\Omega_Y} \mathcal{L}(y_t, h(\phi(\mathbf{x}), t))P(y_t|\mathbf{x})dy_t$. The following derivations show the claim.

$$
\begin{aligned}
\mathcal{L}_{CF} - \mathcal{L}_F &= \int_{\Omega_T} \int_{\Omega_{\mathbf{X}}} l_{h,\phi}(\mathbf{x}, t)P(\mathbf{x})P(t)d\mathbf{x}dt - \int_{\Omega_T} \int_{\Omega_{\mathbf{X}}} l_{h,\phi}(\mathbf{x}, t)P(\mathbf{x}|t)P(t)d\mathbf{x}dt \\
&= \int_{\Omega_T} \int_{\Omega_{\mathbf{X}}} l_{h,\phi}(\mathbf{x}, t) \cdot (P(\mathbf{x})P(t) - P(\mathbf{x}, t))d\mathbf{x}dt \\
&= \int_{\Omega_T} \int_{\Omega_{\mathbf{R}}} l_{h,\phi}(\psi(\mathbf{r}), t) \cdot (P(\psi(\mathbf{r}))P(t) - P(\psi(\mathbf{r}), t))d\psi(\mathbf{r})dt \\
&= \int_{\Omega_T} \int_{\Omega_{\mathbf{R}}} l_{h,\phi}(\psi(\mathbf{r}), t) \cdot (P(\mathbf{r})P(t) - P(\mathbf{r}, t))J_\psi J_\psi^{-1}d\mathbf{r}dt \\
&\leqslant \lambda \cdot \sup_{g \in \Omega_g} \Big| \int_{\Omega_T} \int_{\Omega_{\mathbf{R}}} g(\mathbf{r}, t) \cdot (P(\mathbf{r})P(t) - P(\mathbf{r}, t))d\mathbf{r}dt \Big|.
\end{aligned}
$$

For the third equality, the distribution $P$ over $\Omega_{\mathbf{R}} \times \Omega_T$ can be obtained by the standard change of variables formula, using the determinant of the Jacobian of $\psi(\mathbf{r})$, denoted $J_\psi$ giving $P(\psi(\mathbf{r}), t) = P(\mathbf{r}, t)J_\psi$ (which cancels with the inverse Jacobian that appears after the change of variables in the differential term). The last inequality comes from the assumption that $l_{h,\phi}(\mathbf{x}, t)/\lambda \in \Omega_g$, which is justified and extensively discussed in Shalit et al. (2017).

To prove Thm. 2, we will use the following lemma.

**Lemma 2.** *For convenience, we write $m(t, \mathbf{x}) := \mathbb{E}[Y_t \mid \mathbf{x}]$ and we define its estimate given a prediction function $f : \Omega_{\mathbf{R}} \times \Omega_{\mathbf{X}} \to \Omega_Y$ by $f(t, \mathbf{x})$. If $\mathcal{L}$ is the square loss, it then holds that,*

$$\mathcal{L}_{CF}(t) = \int_{\Omega_{\mathbf{X}}} \int_{\Omega_Y} \mathcal{L}(y_t, h(\phi(\mathbf{x}), t))P(y_t|\mathbf{x})P(\mathbf{x})dy_t d\mathbf{x} \tag{18}$$

$$= \int_{\Omega_{\mathbf{X}}} \int_{\Omega_Y} (y_t - f(\mathbf{x}, t))^2 P(y_t \mid \mathbf{x})P(\mathbf{x})dy_t d\mathbf{x} \tag{19}$$

$$= \int_{\Omega_{\mathbf{X}}} \int_{\Omega_Y} (f(\mathbf{x}, t) - m(\mathbf{x}, t))^2 P(y_t \mid \mathbf{x})P(\mathbf{x})dy_t d\mathbf{x} \tag{20}$$

$$+ \int_{\Omega_{\mathbf{X}}} \int_{\Omega_Y} (m(\mathbf{x}, t) - y_t)^2 P(y_t \mid \mathbf{x})P(\mathbf{x})dy_t d\mathbf{x} \tag{21}$$

$$+ 2 \int_{\Omega_{\mathbf{X}}} \int_{\Omega_Y} (f(\mathbf{x}, t) - m(\mathbf{x}, t))(m(\mathbf{x}, t) - y_t)P(Y_t \mid \mathbf{x})P(\mathbf{x})dy_t d\mathbf{x} \tag{22}$$

$$= \int_{\Omega_{\mathbf{X}}} (f(\mathbf{x}, t) - m(\mathbf{x}, t))^2 P(\mathbf{x})d\mathbf{x} + \sigma_{Y_t}. \tag{23}$$

*The third term in the third equality evaluates to zero because $m(\mathbf{x}, t) := \int_{\Omega_Y} y_t P(y_t \mid \mathbf{x})dy_t$ and we have defined the variance of $Y_t$ with respect to the distribution $P(\mathbf{x})$ as $\sigma_{Y_t} := \int_{\Omega_{\mathbf{X}}} \int_{\Omega_Y} (m(\mathbf{x}, t) - y_t)^2 P(y_t \mid \mathbf{x})P(\mathbf{x})dy_t d\mathbf{x}.$*

**Theorem 2** (Generalization bound for selected treatment tuples $t_1$ and $t_2$). *Let $t_1, t_2 \in \Omega_T$ be two treatment tuples to be compared. Then,*

$$\mathcal{L}_{(t_1,t_2)}/2 \leqslant \mathcal{L}_F(t_1) + \sup_{g \in \Omega_g} \left| \int_{\Omega_{\mathbf{R}}} g(\mathbf{r}) \cdot (P(\mathbf{r}) - P(\mathbf{r} \mid T = t_1)) d\mathbf{r} \right| + \mathcal{L}_F(t_2)$$
$$+ \sup_{g \in \Omega_g} \left| \int_{\Omega_{\mathbf{R}}} g(\mathbf{r}) \cdot (P(\mathbf{r}) - P(\mathbf{r} \mid T = t_2)) d\mathbf{r} \right| - \sigma_{Y_{t_1}} - \sigma_{Y_{t_2}}, \quad (24)$$

*where $\sigma_{Y_{t_1}}$ and $\sigma_{Y_{t_2}}$ stand for the variance of the random variables $Y_{t_1}$ and $Y_{t_2}$, respectively, under the distribution $P(\mathbf{x})$.*

*Proof.*

$$\mathcal{L}_{(t_1,t_2)} = \int_{\Omega_{\mathbf{X}}} (f(t_1, \mathbf{x}) - f(t_2, \mathbf{x}) - m(t_1, \mathbf{x}) + m(t_2, \mathbf{x}))^2 P(\mathbf{x}) d\mathbf{x} \quad (25)$$

$$\leqslant 2 \int_{\Omega_{\mathbf{X}}} (f(t_1, \mathbf{x}) - m(t_1, \mathbf{x}))^2 P(\mathbf{x}) d\mathbf{x} + 2 \int_{\Omega_{\mathbf{X}}} (f(t_2, \mathbf{x}) - m(t_2, \mathbf{x}))^2 P(\mathbf{x}) d\mathbf{x} \quad (26)$$

$$= 2(\mathcal{L}_{CF}(t_1) - \sigma_{Y_{t_1}}) + 2(\mathcal{L}_{CF}(t_2) - \sigma_{Y_{t_2}}) \quad (27)$$

$$\leqslant 2 \left( \mathcal{L}_F(t_1) + \sup_{g \in \Omega_g} \left| \int_{\Omega_{\mathbf{R}}} g(\mathbf{r}) \cdot (P(\mathbf{r}) - P(\mathbf{r}|T = t_1)) d\mathbf{r} \right| - \sigma_{Y_{t_1}} \right) \quad (28)$$

$$+ 2 \left( \mathcal{L}_F(t_2) + \sup_{g \in \Omega_g} \left| \int_{\Omega_{\mathbf{R}}} g(\mathbf{r}) \cdot (P(\mathbf{r}) - P(\mathbf{r}|T = t_2)) d\mathbf{r} \right| - \sigma_{Y_{t_2}} \right). \quad (29)$$

The first inequality holds by the fact that $(a + b)^2 \leqslant 2a^2 + 2b^2$ for any $a, b \in \mathbb{R}$. The second equality holds by Lemma 2 and the last inequality holds by the same arguments used in Theorem 1.

## C EXPERIMENTAL DETAILS

### C.1 DATA GENERATING MECHANISMS

This section describes the data generating mechanisms used in our experiments.

**Synthetic.** We generate synthetic data similar to Nie et al. (2020). With covariates $\mathbf{x} \in \mathbb{R}^6$ all drawn from a uniform distribution between 0 and 1, we generate the continuous dosages and outcomes as follows,

$$\tilde{s}|\mathbf{x} = \frac{10 \sin(\max(x_1, x_2, x_3)) + \max(x_3, x_4, x_5)^3}{1 + (x_1 + x_5)^2} + \sin(0.5 x_3)(1 + \exp(x_4 - 0.5 x_3)) + \quad (30)$$

$$x_3^2 + 2 \sin(x_4) + 2 x_5 - 6.5 + \mathcal{N}(0, 0.25),$$

$$y|\mathbf{x}, s = \cos(2\pi(s - 0.5)) \left( s^2 + \frac{4 \max(x_1, x_6)^3}{1 + 2 x_3^2} \sin(x_4) \right) + \mathcal{N}(0, 0.25), \quad (31)$$

where $s = (1 + \exp(-\tilde{s}))^{-1}$.

**IHDP Continuous.** The IHDP dataset contains 25 covariates with binary treatments and continuous outcomes (Hill, 2011). Disregarding original treatments and outcomes, we use the covariates to generate new continuous dosages and outcomes to test our method. We follow the data generating

procedure of Nie et al. (2020), namely:

$$\tilde{s}|\mathbf{x} = \frac{2x_1}{1+x_2} + \frac{2\max(x_3, x_5, x_6)}{0.2 + \min(x_3, x_5, x_6)} + 2\tanh\left(5\frac{\sum_{i\in I} x_i - c_2}{|I|}\right) - 4 + \mathcal{N}(0, 0.25), \quad (32)$$

$$y|\mathbf{x}, s = \frac{\sin(3\pi s)}{1.2 - s}\left(\tanh\left(5\frac{\sum_{i\in J} x_i - c_1}{|J|}\right) + \exp\left(\frac{0.2(x_1 - x_6)}{0.5 + \min(x_2, x_3, x_5)}\right)\right) + \mathcal{N}(0, 0.25),$$
$$(33)$$

$$c_1 = \mathbb{E}_{p(\mathbf{x})}\left[\frac{\sum_{i\in J} x_i}{|J|}\right], \quad (34)$$

$$c_2 = \mathbb{E}_{p(\mathbf{x})}\left[\frac{\sum_{i\in I} x_i}{|I|}\right], \quad (35)$$

where $s = (1 + \exp(-\tilde{s}))^{-1}$, $I = \{16, 17, 18, 19, 20, 21, 22, 23, 24, 25\}$, and $J = \{4, 7, 8, 9, 10, 11, 12, 13, 14, 15\}$.

**News.** This dataset contains words sampled from 5000 news articles (Newman, 2008). The covariates are word counts. We generated continuous dosage and outcomes by following the data generation method listed in Bica et al. (2020). We first sample three vectors $\mathbf{v}'_i \sim \mathcal{N}(0, 1)$, with $\mathbf{v}_i = \mathbf{v}'_i/||\mathbf{v}'_i||_2$ for $i = 1, 2, 3$. Then, dosages are drawn from a Beta distribution:

$$s \sim \text{Beta}(2, \beta), \quad \beta = \max\left(1, \left|\frac{2\mathbf{x}^T\mathbf{v}_2}{\mathbf{x}^T\mathbf{v}_1}\right|\right). \quad (36)$$

Finally, outcomes are sampled according to:

$$y' = \exp\left(\left|\frac{\mathbf{x}^T\mathbf{v}_2}{\mathbf{x}^T\mathbf{v}_1}\right| - 0.3\right) \quad (37)$$

$$y = 2\left(\max(-2, \min(2, y') + 20\mathbf{x}^T\mathbf{v}_3 * (4(s - 0.5)^2) * \sin\left(\frac{\pi s}{2}\right)\right) + \mathcal{N}(0, 0.25). \quad (38)$$

### C.2 ARCHITECTURES AND TRAINING DETAILS

In both VCNet and DRNet, the representation part of the network $\phi_\eta$ and the prediction heads $h_\theta$ have two layers each, with 50 hidden units and ReLU activations. Following Nie et al. (2020), we use B-spline with degree 2 and knots placed at $\{1/3, 2/3\}$ for VCNet and 5 regression heads for DRNet.

For the MLP model, we use a 4-layers network to represent similar power of approximations to ensure fair comparison. We optimise the networks using Adam (Kingma & Ba, 2014) with a weight decay of 0.005 for regularisation and a batch size of 1000. Learning rate is chosen within the set $\{0.01, 0.005, 0.001, 0.0005, 0.0001, 0.00005\}$ using the procedure outlined in C.3. Each data set is split into a train/validation/test set with ratios 0.6/0.2/0.2. To avoid overfitting, we stop the training if the validation loss did not improve after 50 epochs.

**Propensity score regularization (-PS methods)** In addition to the representation net $\phi : \Omega_\mathbf{X} \to \Omega_\mathbf{R}$ and to the prediction net $h_1 : \Omega_\mathbf{R} \times \Omega_T \to \Omega_Y$, propensity score regularized methods also include a separate head $h_2 : \Omega_\mathbf{R} \to \Omega_T$. Parameters are tuned by minimizing the loss

$$\mathcal{L}_{PS}(\phi, h_1, h_2) = \frac{1}{N}\sum_{n=1}^{N}\left(y^{(n)} - h_1(\phi(\mathbf{x}^{(n)}), t^{(n)})\right)^2 + \alpha \cdot \text{CrossEntropy}\left(h_2(\phi(\mathbf{x}^{(n)})), t^{(n)}\right). \quad (39)$$

In our experiments, $h_2$ is modelled through a softmax layer over a grid of 10 bins. Average treatment effects are then estimated by considering an additional perturbation term following Shi et al. (2019) and Nie et al. (2020). $\alpha$ is treated as a hyperparameter and chosen within the set $\{0.5, 1\}$ using the procedure detailed in C.3. The implementation in practice follows the publicly available code of Nie et al. (2020).

|            | **Synthetic** | **IHDP-continuous** | **News**      |
|------------|---------------|---------------------|---------------|
| DRNet      | 55.82 (0.36)  | 49.82 (0.21)        | 82.91 (1.76)  |
| DRNet-PS   | 55.83 (0.30)  | 50.09 (0.17)        | 83.47 (2.00)  |
| DRNet-TR   | 87.71 (0.22)  | 79.65 (0.58)        | 125.17 (2.77) |
| DRNet-HSIC | 83.28 (1.47)  | 75.86 (1.44)        | 134.47 (1.74) |
| DRNet-WASS | 63.57 (0.42)  | 59.67 (0.30)        | 99.20 (1.64)  |
| VCNet      | 32.73 (0.15)  | 29.71 (0.34)        | 77.51 (0.23)  |
| VCNet-PS   | 32.91 (0.21)  | 29.79 (0.21)        | 76.13 (0.26)  |
| VCNet-TR   | 51.82 (0.24)  | 43.82 (0.17)        | 114.03 (0.30) |
| VCNet-HSIC | 59.43 (0.61)  | 53.88 (0.94)        | 128.23 (0.76) |
| VCNet-WASS | 41.69 (0.29)  | 38.64 (0.29)        | 91.83 (0.52)  |

Table 2: Computational times (in seconds) required for 2000 epochs of training. Averages and standard deviations (within brackets) over 10 runs (5 for News dataset) are reported.

**Targeted regularization (-TR methods)** Methods labeled -TR use the functional targeted regularization approach presented in Nie et al. (2020) which optimizes the loss function

$$\mathcal{L}_{TR}(\phi, h_1, h_2, \epsilon_N) = \mathcal{L}_{PS}(\phi, h_1, h_2) + \frac{\beta}{N} \sum_{n=1}^{N} \left( y^{(n)} - h_1(\phi(\mathbf{x}^{(n)}), t^{(n)}) - \frac{\epsilon_N(t^{(n)})}{h_2(\phi(\mathbf{x}^{(n)}))} \right)^2 \tag{40}$$

where $\epsilon_N(\cdot) = \sum_{j=1}^{J_N} a_j \psi_j(\cdot)$ is modelled through $J_N$ spline basis functions $\psi_j$ of degree 2. The number of basis might change with the sample size $N$. Following Nie et al. (2020), we select the learning rate for $\epsilon_N(\cdot)$, $\beta$ and the number of spline knots within the sets $\{0.001, 0.0001\}$, $\{20, 10, 5\}/\sqrt{N}$ and $\{5, 10, 20\}$, respectively. Again, the implementation in practice follows the publicly available code of Nie et al. (2020).

**IPM regularization (-HSIC and -Wass methods)** IPM regularized methods minimize the proposed loss in Equation (13) in the main body of this paper, where the $\gamma$ is selected within the set $\{10^{i/6}, i = -18, -17, \cdots, 9, 10\}$ using the procedure in C.3. The implementation of Wasserstein distance regulariser follows the one available at `https://github.com/clinicalml/cfrnet/blob/master/cfr/util.py#L166`. HSIC regulariser is computed according to (Greenfeld & Shalit, 2020, Eq. 3), using two RBF kernels with length-scales $\{0.05, 0.1, 0.5, 1, 5, 10, 50, 100, 500\}$.

**Generalised Propensity Score (GPS)** We use our own implementation following Hirano & Imbens (2004).

### C.3 HYPER-PARAMETERS TUNING

We use grid-search to tune the hyper-parameters. Namely, we generate a dataset for each hyperparameters setting, randomly splitting it into a train/test set with a ratio of 0.8/0.2 and we choose the hyperparameters values giving the best MISE test score.

### C.4 RUN TIME COMPARISONS

Table 2 reports the computational time (in seconds) required by the algorithms compared in the experimental section for 2000 training epochs. These results are machine and algorithm- specific but do serve as a relative comparison of run times for different neural network architectures and regularization techniques. In general, Wassertstein IPM regularisation is more computationally efficient than TR and IPM regularisation through HSIC metric.

| | Synthetic | | IHDP-continuous | | News | |
|---|---|---|---|---|---|---|
| | $\sqrt{\text{MISE}}$ | $\sqrt{\text{AMSE}}$ | $\sqrt{\text{MISE}}$ | $\sqrt{\text{AMSE}}$ | $\sqrt{\text{MISE}}$ | $\sqrt{\text{AMSE}}$ |
| GPS | 2.80 (0.51) | 2.75 (0.52) | 4.91 (0.87) | 4.88 (0.88) | - | - |
| MLP | 0.72 (0.09) | 0.62 (0.11) | 0.74 (0.07) | 0.59 (0.05) | 1.05 (0.07) | 0.66 (0.15) |
| DRNet | 0.34 (0.06) | 0.20 (0.04) | 0.60 (0.06) | 0.42 (0.05) | 0.84 (0.04) | 0.37 (0.09) |
| DRNet-PS | 0.43 (0.07) | 0.25 (0.06) | 0.66 (0.15) | 0.43 (0.12) | 0.84 (0.04) | 0.37 (0.09) |
| DRNet-TR | 0.41 (0.03) | 0.17 (0.02) | 2.07 (3.54) | 0.68 (0.70) | 0.82 (0.05) | 0.30 (0.10) |
| DRNet-HSIC | 0.39 (0.06) | 0.22 (0.02) | 0.52 (0.05) | 0.35 (0.06) | 0.87 (0.04) | 0.44 (0.07) |
| **DRNet-Wass** | 0.38 (0.07) | 0.21 (0.02) | **0.51 (0.08)** | 0.34 (0.08) | 0.86 (0.04) | 0.43 (0.07) |
| VCNet | 0.33 (0.03) | 0.13 (0.06) | 0.56 (0.09) | 0.33 (0.09) | 0.85 (0.05) | 0.31 (0.11) |
| VCNet-PS | 0.62 (0.42) | 0.32 (0.30) | 0.59 (0.11) | **0.31 (0.12)** | 0.98 (0.09) | **0.25 (0.13)** |
| VCNet-TR | 0.42 (0.12) | 0.19 (0.11) | 0.64 (0.39) | **0.31 (0.13)** | 0.99 (0.06) | 0.4 (0.08) |
| **VCNet-HSIC** | **0.28 (0.04)** | **0.10 (0.03)** | 0.56 (0.09) | 0.33 (0.08) | 0.87 (0.05) | 0.35 (0.11) |
| **VCNet-Wass** | 0.38 (0.03) | 0.11(0.03) | 0.55 (0.10) | 0.33 (0.08) | **0.81 (0.05)** | 0.29 (0.1) |

Table 3: Average values and standard deviations (within brackets) of $\sqrt{\text{MISE}}$ and $\sqrt{\text{AMSE}}$ across 50 (20) realizations of Synthetic, IHDP-continuous (News) datasets. Bold notation highlights the best-performing algorithm on each dataset.

## D  FURTHER EXPERIMENTS

This section includes experiments comparing performance with respect to Average Mean Squared Error (AMSE), in addition to Mean Integrated Squared Error (MISE),

$$\text{MISE} = \frac{1}{N}\frac{1}{k}\sum_{n=1}^{N}\sum_{w\in\Omega_W}\mathbb{E}\left[\left(y^{(n)}(w,s) - \hat{y}^{(n)}(w,s)\right)^2\right],$$

$$\text{AMSE} = \frac{1}{k}\sum_{w\in\Omega_W}\mathbb{E}\left[\left(\frac{1}{N}\sum_{n=1}^{N}(y^{(n)}(w,s) - \hat{y}^{(n)}(w,s))\right)^2\right],$$

where $y^{(n)}(w,s)$ and $\hat{y}^{(n)}(w,s)$ stand for the true and predicted outcome for individual $n$ given treatment-dosage pairs $(w,s) \in \Omega_T$, and $\mathbb{E}g(s) = \int_{\Omega_S} g(s)P(s)ds$. The AMSE calculates the accuracy of the population level dose response.

As the doubly robust methods get rid of effect modifiers, that are useful for accurate predictions, but have theoretical guarantees for the average treatment effects, we expect these methods to get a better AMSE. On the other hand, as the regularizers proposed in this work provide guarantees on the counterfactual error, we expect models trained with these to achieve a better MISE score.

In Table 3 we show mean performance for both MISE (as in the main body of this paper) and AMSE with the objective to contrast the proposed methods, designed for optimal conditional average counterfactual prediction, i.e. MISE, and doubly-robust methods designed for population average counterfactual prediction, i.e. AMSE. Overall, we note that the proposed regularization technique, using either the HSIC or Wasserstein distances between distributions, is competitive across all datasets and metrics, including AMSE.

Across all datasets, there is a clear trend showing that regularizing for optimal generalization performance in terms of the MISE with HSIC leads to good population average performance as well, as measured by AMSE. Doubly-robust methods (-PS, -TR) are designed for optimality in estimation of the AMSE, and across all datasets they are either optimal or competitive compared to all other algorithms. It is interesting to note that the performance achieved by the proposed regularisation techniques (HSIC, Wasserstein) are very close to the optimum AMSE while, in contrast, doubly robust methods often perform significantly worse in terms of MISE than the optima achieved by HSIC and Wasserstein regularisation. This further confirms the intuition presented in Appendix A: estimating average counterfactuals and individual counterfactuals can require different objectives.

Moreover, in terms of AMSE, it still holds that neural network architectures with better expressiveness to model heterogeneous dose-response curves perform better on all datasets.

