# OpenReview forum: "Generalization bounds and algorithms for estimating the effect of multiple treatments and dosage"
_ICLR.cc/2023/Conference — Submitted to ICLR 2023_

### Official Review · Reviewer_BdyF · 2022-10-17

**Confidence:** 4
**Correctness:** 3
**Technical Novelty And Significance:** 2
**Empirical Novelty And Significance:** 2
**Recommendation:** 5

**Clarity, Quality, Novelty And Reproducibility:**

The paper is very clearly written, some of the details of the experiments are lacking (e.g., the dimension of the intervention variable in the experimental results) but overall the paper is clear.
See notes under strengths and weaknesses about novelty.

**Strength And Weaknesses:**

Strengths:

Because the majority of work in causal literature focuses on binary interventions, this paper represents an important and understudied problem in the field of causality. It is likely to be of interest to a wide audience.

Weaknesses:

The main limitation that I see with this work is that it represents the most straightforward extension of Johansson/Shalit’s work on binary treatments, without addressing the inherent challenges that come with the high dimensionality of the intervention variable. Specifically:

1. One of the core limitations that prevents extensions of methods designed for binary treatment effect estimation the high dimensional treatment setting is that in this setting the overlap assumption is more likely to be violated. This is the kind of challenge that I would expect any paper on high dim treatment to tackle head on, so it was rather disappointing that the authors did not.

2. Another core limitation preventing extensions from binary to categorical extensions (in the framework presented by Johansson/Shalit et al) is the fact that naive extensions would lead to multiple heads and extreme sample splitting (meaning each head is updated by a very small number of samples). However, this is the direction that the authors choose to take. Can the authors comment on how they expect their model to perform as k increases? This is particularly hard to assess given that it seems like all the experiments seem to have been done on datasets with k=1? At least that was my understanding from reading the appendix in Nie et al. Can the authors confirm that none of the experiments were done on k>1?

3. Of note is that those two limitations have been addressed elsewhere (see [Kaddour et al](https://arxiv.org/abs/2106.01939)). Can the authors comment on how their approach compares (both theoretically and empirically) to work by Kaddour et al? It seems to me that Kaddour et al address the two main limitations of high dim/arbitrary type of interventions that this work does not address.






**Summary Of The Paper:**

The authors develop an approach for conditional average treatment effect estimation for interventions that have an arbitrary dimension and type (i.e., could be continuous valued or binary). They do so by extending previous work by Johansson/Shalit on binary treatments.


**Summary Of The Review:**

The paper tackles an important issue but it falls a bit short of suggesting a truly novel approach that deals with challenges that naive extensions of binary treatment effect estimation are prone to.

---

### Official Review · Reviewer_Z4Tv · 2022-10-23

**Confidence:** 3
**Correctness:** 3
**Technical Novelty And Significance:** 2
**Empirical Novelty And Significance:** 2
**Recommendation:** 6

**Clarity, Quality, Novelty And Reproducibility:**

I will put some issues and questions in this section.

## Notations
1. $P(X = x)$ and $P(Y=y | X=x)$, which stands for the probability, seems natural to me only when $\{X,Y\}$ are discrete. I think it's desirable to avoid the notation $Y=y$ or $X=x$ when they are continuous. Or, it should be stated that "$P(x)$" is a *probability mass function or probability density function* instead of "*probabilities or densities*" because the density is not necessarily interpreted as a probability for a continuous variable.
2. In Assumption 1, it should be $T = (W,S)$, not $t = (w,s)$, and $X$ not $x$, because the ignorability is stated for a random variable, not its realization.
3. In Assumption 2, if $T$ is continuous, the positivity assumption should be stated as $P(t | x) > 0$ because the `density' $P(t |x)$ can be over 1 when T is continuous.

## Issues in some results (updated)

1. In Proposition 1, the identification holds with Assumptions {1,2,3}. Without positivity, the identification doesn't hold.



2. **[updated: This issue is addressed by the authors' response]** I don’t understand insightful interpretation of $L_F$ and $L_{CF}$. Specifically, how it seems to me that (when T is discrete and assume the loss function is non-negative)
$$
L_F(t) = \mathbb{E}[L(Y(t), X, T \vert T=t],
$$
and
$$
L_{CF}(t) = \sum_{t' \neq t}\mathbb{E}[L(Y(t),X,T \vert T=t'] P(t') = L_F - L_F(t)P(t) \leq L_F ,
$$
where $L(Y(t), X, T) \coloneqq L(Y(t), h(\phi(X),T))$ a loss function of $\phi, h$ for estimating $Y(t)$, and $L_F \coloneqq \mathbb{E}[L(Y(t), X,T]$. Then, this implies that
$$
L_{CF} \leq L_F.
$$
This generalization bound is tighter than the one proposed in the paper. Therefore, I think finding a representation function minimizing $L_F$ will also bound the counterfactual risk. Given that, what would be an interpretation and meaning of the Theorem 1?

## Questions.
1. Is Corollary 1 a known result? Shouldn't it be the case that $\phi$ is an invertible mapping? To my knowledge, this holds when $\phi$ is a balancing score, including the propensity score. Other than that, I didn't know that Corollary 1 holds.
2. How can one guarantee that a choice of function class g satisfies an assumption in Theorem 1? Do the proposed choices (RKHS and Lipschitz) satisfy the assumption?
3. The paragraph titled "Bias-Variance tradeoff" states that the bounds will be tight if and only if $\phi(X) \perp T$ holds. However, given that $\phi$ is a surjective function (which is one-to-one), can we find a representation function satisfying $\phi(X) \perp T$ while X and T are dependent?

**Details Of Ethics Concerns:**

This paper doesn't have ethical concerns.

**Strength And Weaknesses:**

## Strength
1. This paper is well-written.
2. This paper is self-contained.
3. This paper contains extensive simulation results.

## Weakness
1. Some theories can be more formalized. Details are provided in the next section.
2. Intuitive/insightful explanations about the proposed results are missing.


**Summary Of The Paper:**

This paper provides a generalization bound for a treatment effect on the treated (ETT) with a representation learning function and develops a learning principle based on the proposed bounds.

**Summary Of The Review:**

Even if the paper is well-written, I think the paper needs intuitive and insightful explantions about Theorem 1.

---

### Official Review · Reviewer_zcLw · 2022-10-26

**Confidence:** 4
**Correctness:** 3
**Technical Novelty And Significance:** 2
**Empirical Novelty And Significance:** 2
**Recommendation:** 5

**Clarity, Quality, Novelty And Reproducibility:**

Paper is well written, however the quality suffers due to the lack of attention paid to prior art. Novelty suffers for the same reason.

**Strength And Weaknesses:**

Strengths:
* Counterfactual estimation for general treatment regimes is an important problem
* The authors do a nice job of motivating the loss functino

Weaknesses:
* The paper does not seem to engage with the literature on learning for continuous and multivalued treatments. For example, "Regret Minimization for Causal Inference on Large Treatment Space" from ICML 2021 makes nearly the same observation as this paper and employs HSIC for dependence minimization. Additional examples include "Double Debiased Machine Learning Nonparametric Inference with Continuous Treatments", which addresses the problem of learning continuous treatment effects using machine learning, and the balancing weight literature which explicitly expresses the risk of the causal estimate in terms of dependence between treatment and covariates.

**Summary Of The Paper:**

This paper addresses the problem of representation learning for counterfactual estimation under non-binary treatments. The authors observe that counterfactual error can be reduced by ensuring independence between treatments and covariates. Using this observation the authors propose a loss and show improved empirical results with respect to other proposed learners based on neural networks.

**Summary Of The Review:**

As I stated above, I believe that the novelty and significance of this paper is extremely limited by the lack of contextualization within the larger literature on causal inference for general valued treatments. I would encourage the authors to better contextualize the results of their work and provide comparison against relevant methods from both inside and outside the representation learning community.

---

### Official Review · Reviewer_RujR · 2022-11-02

**Confidence:** 2
**Correctness:** 4
**Technical Novelty And Significance:** 2
**Empirical Novelty And Significance:** 3
**Recommendation:** 5

**Clarity, Quality, Novelty And Reproducibility:**

The paper needs to be proofread for consistency and jargon. Some specific instances include "variance regularization proposal" from previous works which is not explained in the paper. Definition 3 abuses notation of \mathcal{L} by passing \tau_(t_1,t_2)(x) which are not members of \Omega_y. IPM is defined the second time it appears (in (9)) as opposed to the first time it appears in (6).


**Strength And Weaknesses:**

The paper extends the generalization bounds for treatment effect estimation for the multiple treatments with continuous dosage setting. While novel, it doesn't seem like this extension is nontrivial. The bounds are of the same form. The paper would be much stronger if the differences and contributions of the paper are highlighted over just mentioning that they extend the binary treatment results. The experimental validation however, is thorough and the proposed methods and regularizers are better than existing approaches.

**Summary Of The Paper:**

This paper seeks to define appropriate representations of the coviariates for treatment effect estimation in the setting of multiple treatments with continuous dosage parameters. Previous work has looked at this problem in the binary treatment setting. As in previous work, the paper defines a standard ML loss and a counterfactual loss. Bounds for the counterfactual loss are derived in terms of the standard ML loss and the IPM between the joint distribution of the treatment and representation and the product of marginals for a sufficiently expressive function class. The key intuition is to increase predictive accuracy while making the representation as independent as possible of the treatment. Experimentally, different function classes for the IPM and different network architectures are implemented and compared.

**Summary Of The Review:**

While the experimental results are impressive, I think I need to be convinced that this is not just a trivial extension of the binary treatment setting. The paper as it stands doesn't make this case very strongly.

---

### Decision · Program_Chairs · 2023-01-20

**Decision:**

Reject

**Justification For Why Not Higher Score:**

Incremental contributions

**Justification For Why Not Lower Score:**

N/A

**Metareview: Summary, Strengths And Weaknesses:**

Even though the problem is of practical interest, the contributions are too incremental in nature compared to prior work for the ICLR bar.